

# Spatial-temporal characteristics of the oceanic bottom mixed layer in the South China Sea

Wenhu Liu[1,2], Guihua Wang[1,3], Changlin Chen[1,2], and Muping Zhou[4]

[1]Department of Atmospheric and Oceanic Sciences & Institute of Atmospheric Sciences, Fudan University, Shanghai, 200433, China.
[2]Southern Laboratory of Ocean Science and Engineering (Guangdong, Zhuhai), Zhuhai, 519000, China.
[3]CMA- FDU Joint Laboratory of Marine Meteorology, Shanghai, 200433, China.
[4]State Key Laboratory of Satellite Ocean Environment Dynamics, Second Institute of Oceanography, Ministry of Natural Resources, Hangzhou, 310012, China.

*Correspondence to*: Guihua Wang (wanggh@fudan.edu.cn)

**Abstract.** The oceanic bottom mixed layer (BML) has an important role in transporting mass, heat, and momentum between the ocean interior and the bottom boundary. However, the spatial-temporal variability of the BML in the South China Sea (SCS) is not well understood. Using historical hydrological data from 2004 to 2018 and observations from two hydrographic moorings in the SCS, it has been found that the BML in the SCS has significant inhomogeneity. In particular, while the BML is relatively thin and stable over the continental shelf and in deep-sea regions, it is thick and unstable over the northern continental slope. The typical thicknesses of the BML on the continental shelf, continental slope, and deep-sea regions are around 30–60 m, 80–120 m, and 10–50 m, respectively. Statistically, the mean, median, and one standard deviation values of BML thickness over the entire SCS are 73 m, 56 m, and 55 m, respectively. Further analysis reveals that energetic high-frequency dynamic processes, together with steep bottom topography (large slope and roughness), cause strong tidal dissipation and vertical mixing near the bottom over the continental slope, resulting in thicker BMLs there. In contrast, the dynamic processes in the deep ocean are less energetic and low-frequency and the topography is relatively smooth (smaller slope and roughness). Therefore, the tidal dissipation and bottom vertical mixing in the deep-sea regions are weaker, and the BML is relatively thin. These findings may enhance our understanding of the BML dynamics in the SCS and other marginal seas and provide insights that help to improve parameterizations of physical processes in ocean models.

## 1 Introduction

The oceanic bottom mixed layer (BML) is the section of the water column adjacent to the seafloor where active mixing promoted by bottom shear and/or internal wave breaking yields a vertically homogeneous or quasi-homogeneous profile for the seawater properties (Huang et al., 2019; Armi and Millard, 1976). Within the BML, mass, heat, and momentum can cross streamlines, and exchange the physical, chemical, and biological properties between the bottom boundary and the ocean interior (Thorpe, 1988; Trowbridge and Lentz, 2018). The BML is also important for the dissipation of the energy contained



in the large-scale ocean currents (Munk and Wunsch, 1998). Therefore, properly quantifying these exchanges and energy sinks requires a thorough understanding of the nature and behavior of the BML (Peter and Garrett, 2004; de Lavergne et al., 2016).

However, accurately identifying and modeling the BML in the ocean is a challenging task. The typical thickness of BML ($H_{BML}$) in the oceans ranges from a few tens to hundreds of meters (Armi and Millard, 1976; Lozovatsky et al., 2008; Huang et al., 2019). Unfortunately, the $H_{BML}$ is still not well captured and simulated by most of the current ocean general circulation models due to the coarse vertical resolutions typically used near the bottom (typical 100–200 m at the nearest model layer) and/or the inadequate parameterizations used for the bottom boundary layer (Peter and Garrett, 2004; Fox-Kemper et al.,

2019). There remain significant gaps in our understanding of the basic structure of the BML and its spatial-temporal variability in both observations and modeling (de Lavergne et al., 2016).

The formation of the BML is associated with mixing processes at the ocean's bottom boundary that are closely linked with active dynamic processes and topographic features (Polzin and McDougall, 2022). Observations show that boundary mixing is usually enhanced over steeply sloping and rough topography (Moum et al., 2002; Polzin et al., 1997; St. Laurent et al.,

2001; Ledwell et al., 2000). This enhanced mixing occurs not only within the BML but also in a stratified region outside it. Consequently, the structure of the BML may have a more complicated structure over steeply sloping and/or rough topography (Armi and Millard, 1976; Polzin et al., 1997). In addition, stable stratification inhibits vertical mixing and instability processes and decreases the $H_{BML}$ (Weatherly and Martin, 1978). Besides, if there exists geothermal heating through the ocean bottom, it can also greatly change the $H_{BML}$ through thermal diffusion and/or convection processes

(Banyte et al., 2018). Thus, the dynamic factors that determine the structure of the BML can be complex.

The South China Sea (SCS), which contains a deep-sea basin as well as a wide continental slope, is the largest marginal sea in the western Pacific Ocean (Figure 1a). Enhanced turbulent mixing in the SCS is believed to play a key role in driving the water exchange between the SCS and the Pacific and promoting upwelling from the deep SCS basin (Lu et al., 2021; Tian et al., 2009; Quan and Xue, 2019), however where and how the upwelling occurs is still debated. Therefore, properly assessing

the bottom mixing and associated dynamic processes will undoubtedly help us to better understand and quantify the upwelling from the deep SCS.

A significant amount of work has been devoted to studying the mixing in the SCS, but only a few efforts have focused on mixing in the deep ocean. Tian et al. (2009) reported the distribution of turbulent mixing along a transect from the northern SCS to the Pacific (117° E–130° E) and found that diapycnal mixing in the SCS was enhanced with a magnitude that was

two orders larger than in the Pacific. Yang et al. (2016) examined the three-dimensional distribution of turbulent mixing in the SCS in hydrographic observations from 2005 to 2012 and found that turbulent mixing was generally intensified in the near-bottom waters. In addition, they found that the regions near the Luzon Strait and around the Zhongsha Island chain were the two mixing hotspots of the SCS. Recently, Lu et al. (2021) used the hydrographic observations in the SCS from 2004 to 2020 to explore the spatial characteristics of turbulent mixing from the surface to 1500 m. They found that the

strongest mixing occurred in the Luzon Strait and Dongsha Plateau regions and that the mixing on the continental slope was





considerably stronger than that in the Xisha and Nansha regions. These mixing features in the SCS were also seen in numerical model simulations (Wang et al., 2016). Although some progress has been made in understanding turbulent mixing in the deep SCS in recent years, the spatial-temporal characteristic of the BML is still unclear due to the lack of observations. Over the past two decades, more than 2000 CTD profiles have been collected from the SCS Open cruises and other programs.

In particular, full-depth CTD observations have increased rapidly in recent years (Figure 1b), which makes it possible for us to explore the structure of BML in the SCS in more detail than before. This study will explore the basic structure of BML in the SCS from the historical hydrological data and mooring observations.

**Figure 1: (a) The map of the South China Sea showing the locations of historical observations. The dots are the CTD cast stations**

**and their color indicates the height of the raw data above the bottom. The red stars indicate the locations of the two mooring stations; The CTD observational frequency statistics are displayed by the (b) year, (c) depth, and (d) month.**



## 2 Data and method

### 2.1 CTD data

The historical hydrographic data collected by in situ observations from the SCS open cruises and other programs during the
past 15 years (2004–2018) are used in this study. These data were obtained from a SBE-911plus conductivity temperature
depth (CTD) system using frequencies between 8–24 Hz. After pre- and post-cruise calibrations, the accuracies of the CTD
sensors are 0.0003 S m⁻¹ for salinity and 0.0018℃ for temperature. To get as complete a vertical profile of the BML as
possible without damaging the CTD sensors by colliding with the bottom, acoustic altimeters were used in the more recent
cruises to monitor the distance of the sensors to the bottom. In this study, only the downcast data are used and the
observations less than 50 m from the bottom are discarded because those data do not accurately capture the structure of the
BML.

The raw CTD data quality control applied the following criteria: (i) remove CTD profiles where the original information
about the station position and/or water depth is missing or incorrect; (ii) because only deeper locations are considered in this
study, stations with depths less than 100 m are excluded; (iii) down sample the raw vertical high-resolution to 1 m and apply
a 5 m running mean filter to smooth the data. Applying these criteria and after validation, only 514 CTD profiles remained.
Nevertheless, these selected CTD data records covered nearly the entire SCS north of 13°N and provided more than
adequate coverage of the northern continental slope region of the SCS (Figure 1a). While the majority of the selected CTD
profiles were located in depths shallower than 500 m, there were 117 CTD profiles at locations deeper than 1000 m (Figure
1c). Most of those profiles were collected from August and September (Figure 1d) because most of the routine SCS
open cruises were conducted in those two months.

### 2.2 Mooring observations

To observe the variations of the BML in the SCS, two bottom anchored moorings were deployed at two sites in the SCS
(Figure 1a). One was deployed on the northern continental slope of the SCS and the other was in the deep basin in the
western SCS, where a previous study had confirmed the existence of a strong deep western boundary current (Zhou et al.,
2020). To obtain simultaneous observations, these two moorings were both deployed in August 2017 and recovered in
September 2018, collecting a 14-month long time series for use in this study. The moorings had seven RBR-TDs and seven
SBE 37 CTDs measuring temperature and pressure near the bottom at M1 and M2, respectively. The accuracies of the RBR-
TDs were ±0.002 ℃ for temperature and ±0.05 % over the full-scale range for pressure. The accuracies of the SBE 37 CTDs
were ±0.002 ℃ for temperature and ±0.1 % over the full-scale range for pressure. The design and configuration of the
moorings used in this study are shown in Table 1. Since there were no salinity measurements on the M1 mooring, the present
study only examines the variation of temperature in the near-bottom regions at the two sites. To remove the influence of tidal
variability from the raw temperature data, a 72-h low-pass filter was used to remove the inertial, tidal, and other high-
frequency signals (Godin, 1966). All data were averaged over an hourly interval.



**Table 1: Experimental mooring design and configuration.**

| Mooring | Location | Depth (m) | Period | Instrument | Design installation depth above the seafloor (m) | Sampling (s) |
|---|---|---|---|---|---|---|
| M1 | 116º 01'E 19º 24' N | 2368 | Aug 6, 2017, to Sep 24, 2018 | RBR-TD | 140/120/100/80/60/40/20 | 600 |
| M2 | 115º 24'E 16º 24' N | 4149 | Aug 1, 2017, to Sep 20, 2018 | SBE37 | 300/250/200/80/60/30/15 | 600 |

**2.3 The other dataset**

To help analyse the potential influence of tidal dissipation on the distribution of $H_{BML}$, a two-dimensional map of the internal tidal dissipation dataset was also used in this study. This dataset uses the framework developed by Eden and Olbers (2014) that breaks down the two-dimensional maps of the constructed climatological low mode dissipation into four dissipative

processes and combines the estimated low-mode dissipation with the local dissipation of higher modes to obtain a horizontal map of the total internal tide dissipation. Note that the tidal energy sinks were decomposed into five process contributions in this dataset, but only the total energy sinks were used in this study. Detailed information on the dataset and documentation can be found in de Lavergne et al. (2019).

**2.4 Identifying the thickness of BML**

In this study, a relative variance method is used to identify the $H_{BML}$ in the SCS. This method is based on the ratio between the standard deviation and the maximum variation of the temperature, salinity, or density profiles above the sea bed; the depth of the minimum relative variance is defined as the top of BML (Huang et al., 2018a). Compared to the commonly used methods, such as the threshold, curvature, and maximum angle methods (Lozovatsky et al., 2008; Lorbacher et al., 2006; Chu and Fan, 2011), the relative variance method is an objective method that determines the $H_{BML}$ that is less dependent on

the any given criteria (Huang et al., 2018b; Huang et al., 2018a). Although the relative variance method was first proposed to identify the surface mixed layer, its performance in determining the $H_{BML}$ is also superior to other available methods (Huang et al., 2019; Huang et al., 2018a). A detailed description of the method and implementation can be found in Huang et al. (2018a).

We use the relative variance method separately on the profiles of potential temperature, salinity, and potential density to

obtain three estimates of the $H_{BML}$ in each CTD cast. The quality index (QI) defined in Lorbacher et al. (2006) was used to evaluate the quality of the estimate of $H_{BML,}$ and profiles with QI <0.5 were discarded. It should be noted that there may exist real differences between the three $H_{BML}$ estimates, and the value with higher QI is adopted as the observed $H_{BML}$. In addition to the formal objective analysis of $H_{BML}$, all profiles used in this study were visually inspected to detect possible errors due to





contaminated samples or accidental spikes. Figure 2 shows an example of temperature and salinity profiles collected at
117.06 °E, 21.45°N near the Dongsha Islands, where a well-mixed layer clearly exists in the near-bottom zone with an $H_{BML}$
of about 100 m.

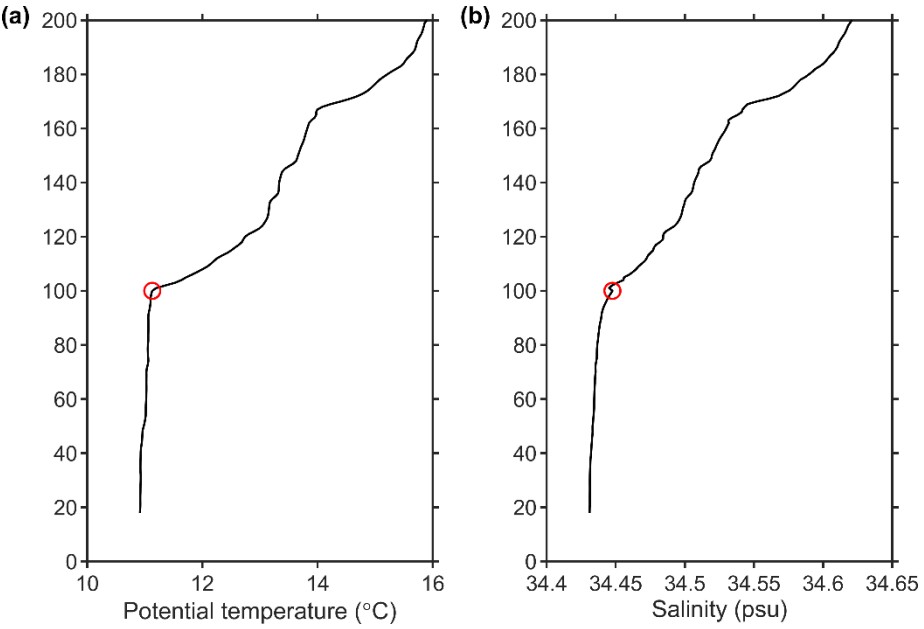

**Figure 2: An example of potential temperature (a) and salinity (b) profiles collected near the Dongsha Islands (117.06° E, 21.45° N), with the total water depth of 403 m. The thickness of the bottom mixed layer is determined to be about 100 m (marked by red**
**circles).**

**2.5 The topographic slope and topographic ruggedness**

To investigate the distribution of $H_{BML}$ in the SCS and its sensitivity to the ocean topography, two main aspects representing
the topographic effects are considered. One is the topographic slope angle ($\theta$) and the other is the topographic ruggedness.
Ruggedness, which measures the degree of irregularity of the topography, is defined by the topographic ruggedness index
(*TRI*), defined by (Riley et al., 1999):

$$TRI = \left[\sum\left(x_{ij} - x_{00}\right)^2\right]^{1/2},$$    (1)

where $i$ and $j$ are the zonal and meridional grid numbers in the specified domain, respectively, and $x_{ij}$ is the elevation of each
neighbor cell relative to the center point cell, $x_{00}$. The *TRI* presents the sum of changes in elevation between a grid cell and
its neighboring cells and is equivalent to the standard deviation in two dimensions (Riley et al., 1999). In this study, we use
the ETOPO1 ocean topography database to calculate the topographic slope and topographic ruggedness on the same
1°×1°spacial domain as that of the bin averaged distribution of $H_{BML}$.



## 3 The observed $H_{BML}$ in the SCS

The $H_{BML}$ computed from the 514 CTD profiles (Figure 3) range from 4–255 m, with the mean, median, and one standard deviation being 73 m, 56 m, and 55 m, respectively. The median value is somewhat thicker than the median value (47 m) in

the global ocean estimated with full-depth CTD data from the World Ocean Circulation Experiment program (Huang et al., 2019). The probability density distribution of $H_{BML}$ (Figure 3) demonstrates that 45% of $H_{BML}$ are in the range of 20~80 m, and 27% of $H_{BML}$ are thicker than 100 m, with a positive skewness (1.25).

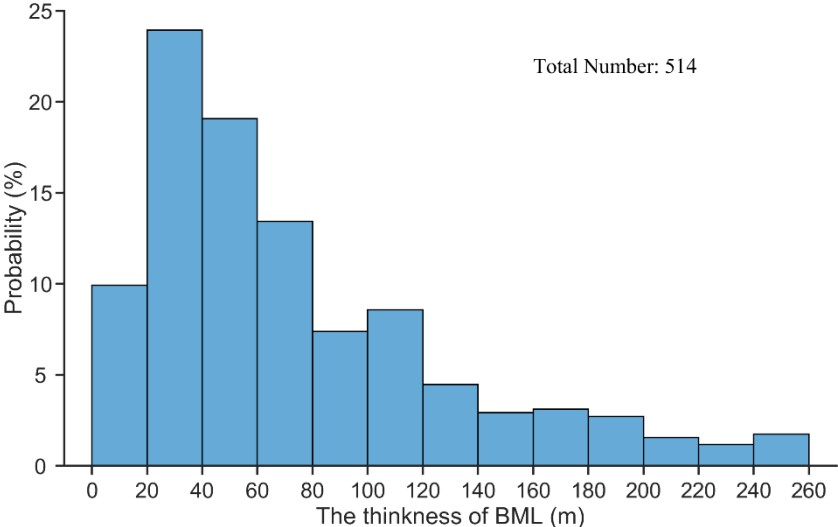

**Figure 3: The probability density distribution of BML thickness in the SCS.**

The thickest averaged $H_{BML}$ (133m), with a standard deviation of 72 m, occurs at a depth of ~700 m (Figure 4a). The correlation between $H_{BML}$ and water depth is positive for water depths shallower than 700 m, and negative for water depths deeper than 700 m. These relationships are quite different from the global distribution presented by Huang et al. (2019), where the $H_{BML}$ increased exponentially for water depths deeper than 1000 m (the statistics for stations shallower than 1000 m were not presented and stations shallower than 500 m were not considered in their study). These differences between the

SCS and the open ocean suggest that the distribution of $H_{BML}$ in the SCS may be regulated by more local dynamic factors rather than the general factors in the open ocean.

To obtain a straightforward statistical relationship between $H_{BML}$ and water depth, we calculate the ratio ($R_{H/D}$) between the $H_{BML}$ and the total water depth ($D$) to evaluate how the $H_{BML}$ varies as a function of depth (Figure 4b). It is interesting to see the $R_{H/D}$ decreases roughly exponentially with water depth following the least-squares best fit curve:

$$R_{H/D} = 0.4007\exp\left(-0.0012D\right) . \tag{2}$$

The mean ratio ranges from 10–50% for depths between ~100–700 m, between 5–10% for depths of ~1000 m, and less than 2% in water deeper than 3000 m. All these values are slightly higher than the results estimated in the North Atlantic

 

(Lozovatsky et al., 2008; Lozovatsky and Shapovalov, 2012), suggesting that bottom mixing is more active in the SCS than in the open ocean.

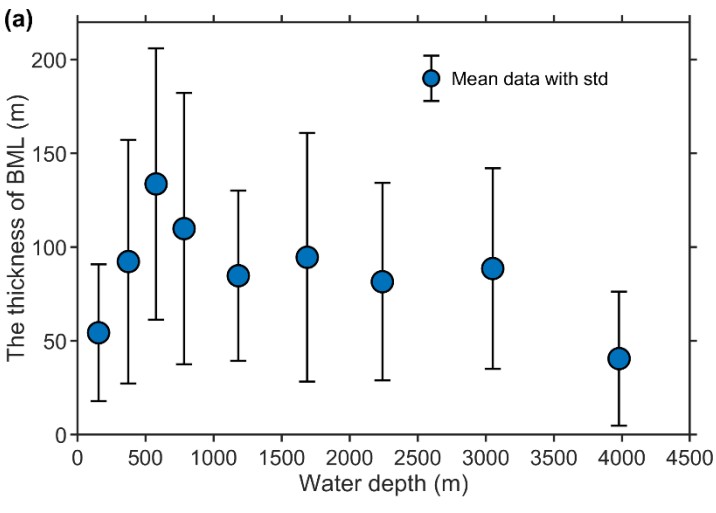

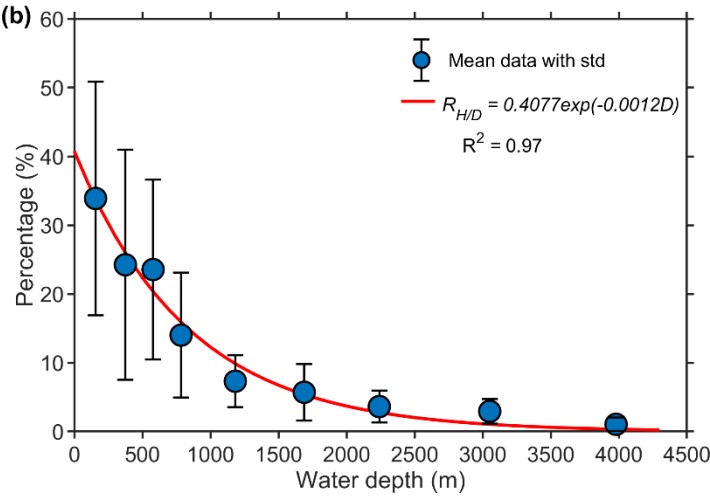

**Figure 4: (a) The distribution of mean BML thickness as a function of the ocean depth; (b) the percentage of BML to the ocean depth as a function of the ocean depth with the least-squares fit superimposed (red curve).**

To investigate the horizontal distribution of $H_{BML}$ in the SCS, we project the geographic scatter data in the 1°×1° bins and calculate the mean values of $H_{BML}$ in each grid cell (Figure 5). The results show that the $H_{BML}$ is thicker (>100 m) on the northern continental slope of SCS, especially in the regions adjacent to the west Luzon Strait and Dongsha Islands, where previous studies have suggested that bottom mixing is enhanced (Lu et al., 2021; Tian et al., 2009; Yang et al., 2016). In contrast, the $H_{BML}$ is thinner over both the continental shelves (around 30–60 m) and the deep-sea regions (around 10–50 m). Despite the limited data in each bin, the variations in the $H_{BML}$ (calculated by using at least 5 data in the bin) were relatively large on the northern continental slope (not shown), indicating a relatively unstable $H_{BML}$ over the continental slope regions.







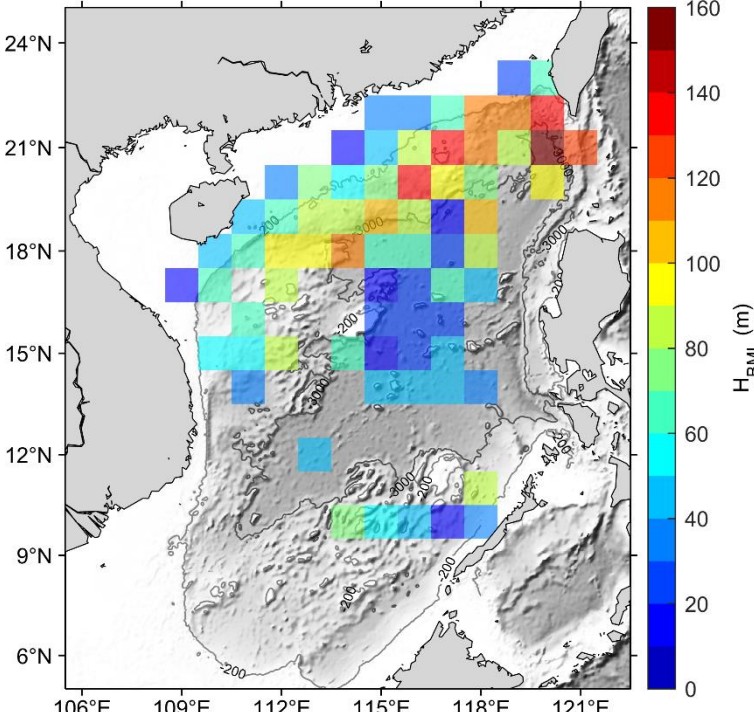

**Figure 5: The horizontal distribution of mean BML thickness averaged in 1°×1° bins.**

Two deep moorings were deployed to measure the variability of the near-bottom potential temperature over the continental slope (M1) and deep-sea (M2) regions. Figure 6 shows the vertical distribution of bottom potential temperature and its corresponding gradient. Although the differences in the potential temperature profiles between M1 and M2 are significant, quasi-homogeneous layer structures in the near-bottom zone can be clearly seen. During most of the observation period, the homogeneous layer is much thicker at the M1 mooring than that at the M2 mooring. The height of the homogeneous layer appears roughly between 100–120 m above the bottom at M1 and 40–60 m above the bottom at M2. This confirms the earlier observation that the $H_{BML}$ over the continental slope is thicker than it is in the deep ocean.




**Figure 6: Mooring observations of the near-bottom (a) potential temperature variations and (b) the corresponding vertical temperature gradients at M1; (c) and (d) are the same as (a) and (b) but for the M2 station. The black lines in panels (b) and (d) indicate the contour line of -4×10⁻⁴ °C m⁻¹ and -0.25×10⁻⁴ °C m⁻¹, respectively.**

To further examine the differences in the $H_{BML}$ distribution between the continental slope (M1) and the deep-sea (M2) regions, an Empirical Orthogonal Function (EOF) analysis (Thomson and Emery, 2014) was performed to compare the dominant vertical structure of the potential temperature gradients. The results show that there exist maximum gradients in both of the first EOF mode of the potential temperature gradients (Figures 7a and 7b). The height where the maximum gradients occurred was 115 m and 50 m at M1 and M2, respectively, supporting the existence of the quasi-homogeneous structure exhibited in Figure 6. The percent of the variance explained by the first EOF is 81.7 % for M1 and 75.1 % for M2,





suggesting that the feature of thicker $H_{BML}$ on the continental slope and thinner $H_{BML}$ in the deep-sea regions is a dominant

feature.

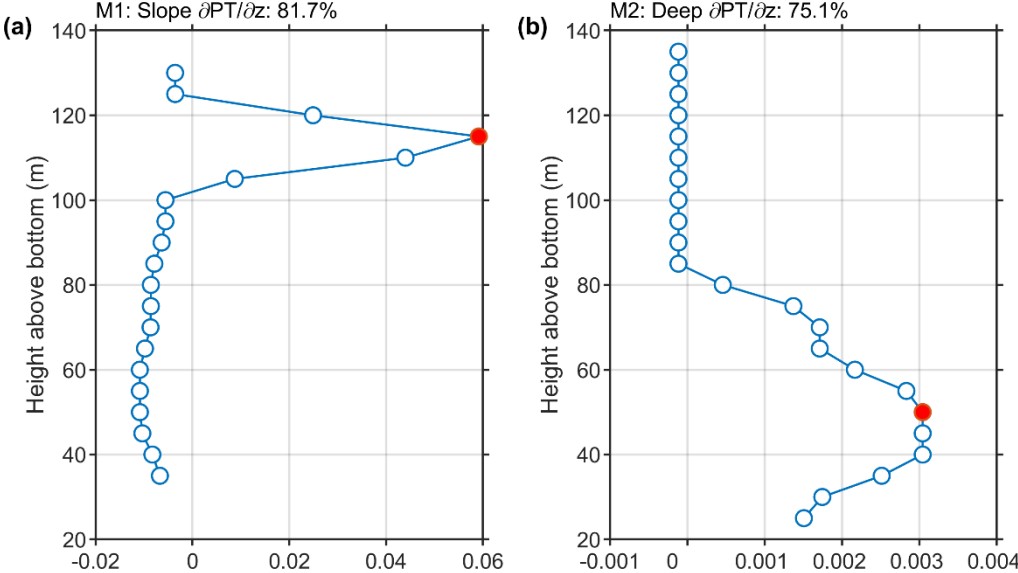

**Figure 7: The vertical distribution of the first-EOF mode of the potential temperature gradient at (a) M1 and (b) M2. The red dots indicate the location of the maximum variations at M1 and M2. The percent of the variance explained by the first EOF at each mooring is given at the top of each panel.**

**4 The potential formation mechanisms of the BML in the SCS**

The fluctuations of bottom temperature are possibly linked with bottom currents that influence the structure of the BML. Spectral analysis of the bottom potential temperatures shows that variability over the continental slope was dominated by the internal tidal and near-inertial signals, while only low-frequency oscillations (~60 days) were significant in the deep ocean (Figure 8). It is suggested that the differences in the $H_{BML}$ between the continental slope and deep ocean may be caused by

different dynamical processes. In the SCS, the internal tides have been shown to be widely distributed on the continental slope both from observations and in model simulations (Wang et al., 2016; Alford et al., 2015) with most of those signals emanating from the Luzon Strait (Zhao, 2014; Alford et al., 2015). However, the near-inertial signals are likely injected into the upper ocean by typhoon processes (Xu et al., 2013) or by other upper ocean mechanisms (Alford et al., 2016). The dominance of these two signals suggests that the internal tidal and near-inertial motions may play the primary roles in the

strong mixing along continental shelves and slopes (Lu et al., 2021; Wang et al., 2016; Yang et al., 2016). For the deep ocean, the low-frequency oscillations (30–120 days) are induced by the topographic Rossby waves or the deep ocean eddies, which has been shown in recent observational and modeling studies (Zheng et al., 2021; Zhou et al., 2017; Zhou et al., 2020; Quan et al., 2021).





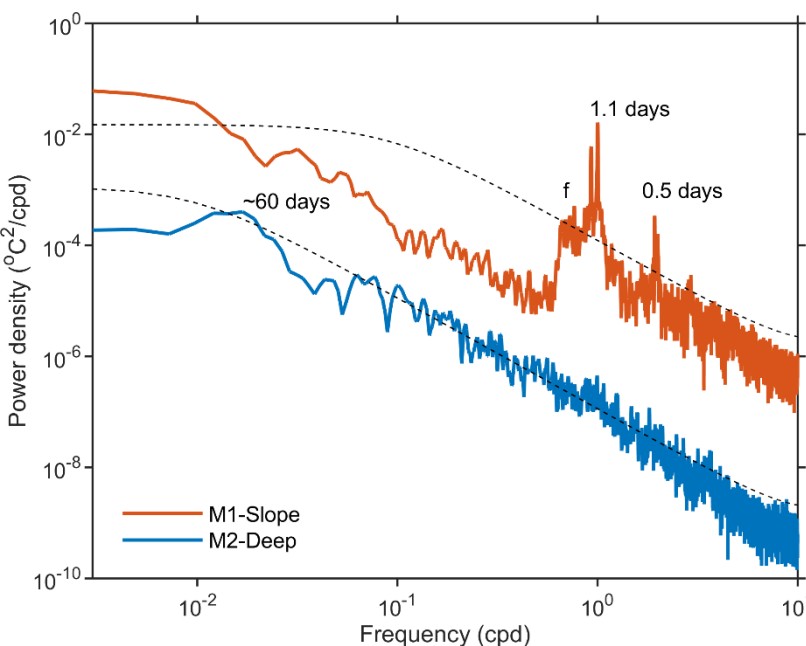

**Figure 8: Power spectrum of the near-bottom potential temperature at the M1 (continental slope) and M2 (deep ocean) mooring sites. The thin lines show the 95% significance level.**

Vertical eddy diffusion is an important process modifying mixing in the two regions. Assuming that vertical advection is balanced mainly by vertical diffusion, the momentum equation of temperature excluding the source and sink terms, becomes (Cushman-Roisin and Beckers, 2011)

$$\frac{\partial T}{\partial t} + w\frac{\partial T}{\partial z} = A_z \frac{\partial^2 T}{\partial z^2},\tag{3}$$

where $T$ is the potential temperature, $t$ is time, z is the vertical coordinate (positive upward), $w$ and $A_z$ are the vertical velocity and vertical eddy diffusion coefficient, respectively. In Eq. (3), the three terms of $\partial T/\partial t$, $\partial T/\partial z$, and $\partial^2 T/\partial z^2$ can be estimated from the mooring observations, and the unknown values of $w$ and $A_z$ can be estimated from a set of linear equations using the least-square fitting method (Thomson and Emery, 2014).

A result of this analysis, the estimated average eddy diffusion coefficient varies between $1.0\times10^{-6}$ to $5.3\times10^{-3}$ m$^2$ s$^{-1}$ at M1 and $2.3\times10^{-7}$ to $5\times10^{-3}$ m$^2$ s$^{-1}$ at M2. These ranges are in agreement with practical observations estimated from the Thorpe-scale method and direct observations in previous studies (Lu et al., 2021; Shang et al., 2017; Tian et al., 2009; Yang et al., 2016). The estimated mean eddy diffusion coefficients were $1.3\times10^{-3}$ m s$^{-1}$ and $8.4\times10^{-4}$ m$^2$ s$^{-1}$ at M1 and M2, respectively. This suggests that the bottom vertical mixing over the continental slope is stronger than that in the deep ocean, which may explain why the $H_{BML}$ over the continental slope is thicker than in the deep ocean.





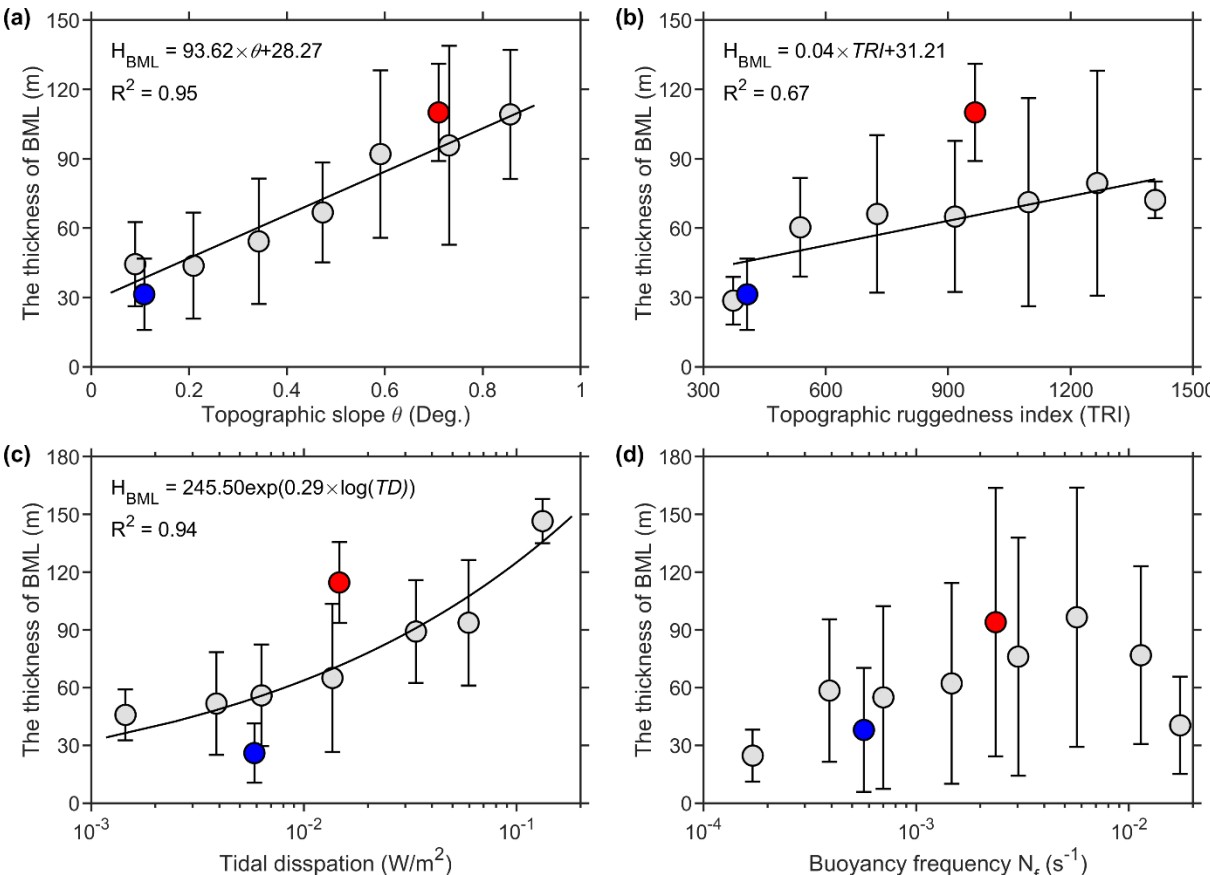

**Figure 9: The distribution of H$_{BML}$ as a function of (a) topographic slope ($\theta$), (b) topographic ruggedness index (*TRI*), (c) internal tidal dissipation (*TD*), and (d) buoyancy frequency (*N$_f$*). Gray dots and error bars denote the average values with one standard deviation. Blue and red dots indicate the mean value near the continental slope (M1) and deep-sea regions (M2), respectively. The best fits are plotted as a black curve in panels (a-c). Note that the averaged buoyancy frequency in panel (d) was estimated within the BML with the thickness of BML determined by Eq. (2).**

Using CTD observations, we analyse the other potential influence factors that contribute to the distribution differences between the two regions. Specifically, we consider the roles of topography, internal dissipation, and density stratification in the distribution of H$_{BML}$. The results in Figures 9a-c show that the H$_{BML}$ increases with increasing topographic slope, topographic ruggedness, and internal tidal dissipation. The typical values of the topographic slope, topographic ruggedness, and internal tidal dissipation are all larger on the continental slope than in the deep ocean and likely all contribute to the thicker H$_{BML}$ over the continental slope. Figure 9d shows the H$_{BML}$ as a function of the buoyancy frequency estimated within the BML. These result shows no monotonic relationship between the H$_{BML}$ and the buoyancy frequency as suggested by Huang et al. (2019), which showed that the H$_{BML}$ tends to be thinner with stronger stratification. However, our results suggest that the stratification may not be important for determining the difference in the BML between the continental slope and the deep ocean. In other words, the dominant factors controlling the distribution of H$_{BML}$ in the SCS are dynamic rather than thermodynamic.



Taken together, we conclude that the dynamic processes over the northern continental slope controlling the BML were the energetic, high-frequency forcing, together with the large slope and steep topography, which combine to cause strong tidal

energy dissipation and vertical mixing near the bottom in these regions. As a result, the BML on the northern continental slope is relatively thick. Conversely, in the deep-sea regions, the dynamic processes are low-frequency, and the topographic roughness and slope are relatively smooth and gentle, so that the tidal energy dissipation and bottom vertical mixing are considerably weaker, resulting in a relatively thin BML in the deep-sea regions.

## 5 Summary and Discussion

In this study, we combined historical hydrological data and observations from two in situ moorings to investigate the spatial-temporal characteristics of the BML in the SCS. In general, the $H_{BML}$ is thicker over the northern continental slope, especially in the region to the west of the Luzon Strait and Dongsha Islands, where the median $H_{BML}$ is thicker than 100 m. In contrast, the $H_{BML}$ is relatively thin over the continental shelves and in deep-sea regions, with median thicknesses of around 30-60 m and 10-50 m, respectively. The values for the mean, median, and standard deviation of $H_{BML}$ in these regions

were 73 m, 56 m, and 55m, respectively. Further analysis revealed that the differences in the $H_{BML}$ between the northern continental slope and deep ocean are due to the different dynamic processes, topographic features, tidal dissipation, and bottom vertical mixing between these two regions. Specifically, the high-frequency energetic dynamic processes and steep topography (large topographic slope and roughness), cause stronger tidal dissipation and bottom vertical mixing over the continental slope, leading to a thicker BML there. Conversely, in the deep ocean, the dynamic processes are low-frequency

and lower energy, and the topography is relatively smooth (small topographic slope and roughness), leading to relatively weak tidal dissipation and vertical mixing near the bottom, and resulting in a thinner BML in the deep-sea regions.

To further explore the distribution of $H_{BML}$ over the entire SCS, we derived a statistical relationship (shown in Figure 9c) between the thickness of $H_{BML}$ and the strength of tidal dissipation. The results show that the thickest estimated $H_{BML}$ appears to the west of the Luzon Strait, the northern slope (especially the northeast continental slope), and surrounding

islands/seamounts. The mean values of $H_{BML}$ in these regions were exceeded 100 m, with the largest $H_{BML}$ over the northern continental slope also apparent in the average thicknesses for each latitude (Figure 10). Both the magnitude and the spatial pattern of $H_{BML}$ are in good agreement with the observed values from the CTD profiles (see Figure 5&Figure 10), suggesting that the tidal dissipation is a dominant factor useful for predicting $H_{BML}$. In addition, according to de Lavergne et al. (2019), several mechanisms contribute to the tidal dissipation which may have different effects on the distribution of $H_{BML}$.

Developing a better assessment of the relative contributions of the different tidal dissipation mechanisms will be important for improving the prediction of the distribution of $H_{BML}$ in the SCS. This would be an important goal of future work in this area.



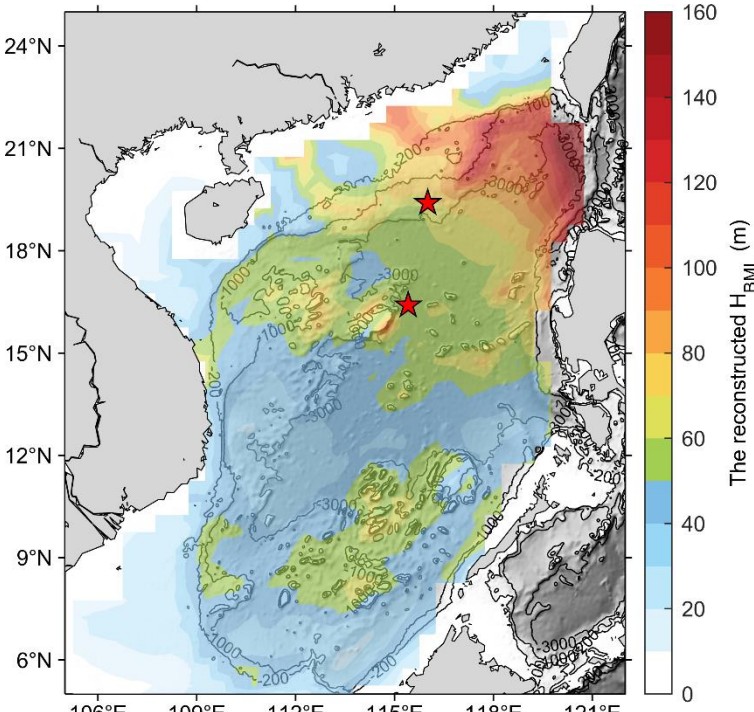

**Figure 10: The spatial distribution of the estimated $H_{BML}$ in the SCS. Two red stars indicate the mooring stations.**


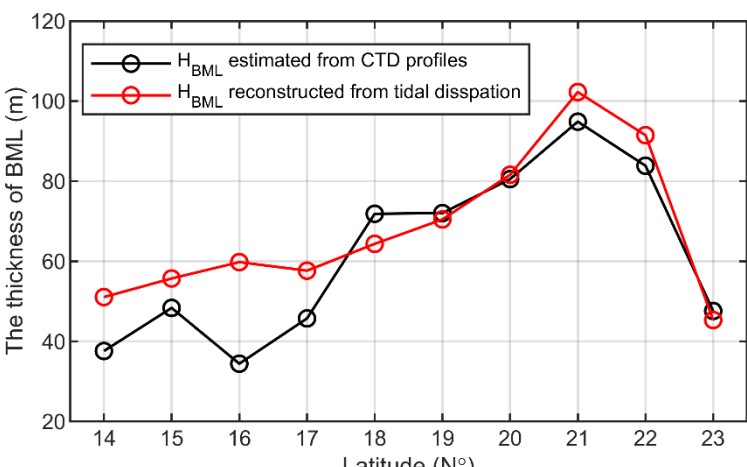

**Figure 11: Comparison of the latitudinal distribution of $H_{BML}$ estimated from the CTD profilers and reconstructed values.**

Our aim in this study was to investigate $H_{BML}$ in the SCS from observations. The result of this study hopefully will be useful for improving our understanding of the dynamics of oceanic BML in the SCS and may provide observational evidence to
refine the parameterization of BML in ocean circulation models. However, the improving our knowledge of the fine structure and obtaining a longer observational record of BML is still urgently needs for future work.



*Data availability*. The topography data are available at: doi:10.7289/V5C8276M. Historical hydrographic data are available at: http://ocean.geodata.cn/data/dataresource.html. The mooring data used in this study are available upon request from the authors.


*Author contributions*. WL collected the observational data, performed the analysis, and drafted the manuscript. GW initiated the idea of the study. All authors contributed to the interpretation of results and editing of the manuscript.

*Acknowledgements*. The authors thank all of the scientists and staff members on the research vessels Experiment 1,
Experiment 3, Jia Geng, Dongfanghong 2, and Dongfanghong 3 for their assistance with the Open Research Cruise of the South China Sea supported by NSFC Shiptime Sharing Projects from 2004 to 2018. The authors are also grateful for the assistance of the Ocean Observation Team from Ocean University of China for the deployment and recovery of the moorings. This work is supported by the National Natural Science Foundation of China (42030405).

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
