# Peer review of "Spatial-temporal characteristics of the oceanic bottom mixed layer in the South China Sea"

_EGUsphere, 2022_

## Author Comment (AC1)

**Reply to reviews of "Spatial-temporal characteristics of the oceanic bottom mixed layer in the South China Sea"**

**Reply to reviewer #1:**

We would like to thank you for your careful reading, helpful comments, and constructive suggestions, which have helped significantly improve the manuscript.

**General Comments:**

This manuscript examines spatio-temporal variations of the BML thickness in the SCS by using accumulated full-depth CTD and two mooring data. It is pointed out that thick/unsteady BML tends to be observed in the highly energetic northern continental slope area while thin/steady BML in the moderately energetic deep-sea area. Such BML variations are attributed to the strength of tidal dissipation purely on the basis of the statistical analysis without being discussed in the light of the most relevant bottom Ekman layer dynamics, which is not acceptable to me. Intuitively, observed thick BML in the highly turbulent area is not surprising, given that the classical Ekman layer theory (albeit in a simplified condition) yields the Ekman layer thickness in terms of the turbulent eddy viscosity. However, it remains to be seen whether existing bottom Ekman layer theories can quantitatively capture observed variations of the BML thickness in the SCS, which, in my opinion, should be rather examined in this study. There are several complications inherent to the bottom Ekman layer dynamics, such as effects of bottom slope and nonconstant eddy viscosity (e.g., Garrett et al. 1993; Muller and Garrett 2004), which may affect the observed results. In addition, observed temporal BML variations seems interesting, whose physical interpretation should be explored much more.

Overall, I cannot support the publication of this manuscript because of the complete lack of discussion of the most relevant bottom Ekman layer dynamics.

Response: We gratefully appreciate your valuable comments. Considering the Reviewer's concerns, we have added specific discussions about bottom Ekman layer dynamics before the mechanism analysis, and narrowed our discussion focus on the BML differences between the

northern continental slope and the deep-sea regions. We have modified the corresponding contents in the revised manuscript.

Although there remains a debate about whether the BML can be treated as the classical Ekman layer, we agree that the nature of the BML is primarily governed by the dynamics of the flow above the seafloor, and this is the bottom Ekman layer dynamics. Our start point of mechanism analysis is based on the Ekman layer dynamics. Because there are only CTD observations used in this study, especially with the lack of the bottom current measurements, we are not able to fully address the formation mechanisms of the BML from perspective of the bottom Ekman dynamics. Nevertheless, our mechanism analysis is based on the bottom Ekman layer dynamics. For example, we explored the flow-induced spectral information from bottom temperature and explored the bottom mixing information from the topographic features and tidal dissipations.

The second concern is about the temporal BML variations and its physical interpretation. Again, there is no enough observational data for us to fully analyze the mechanisms of the temporal variations of the BML. Our mooring observations show that BML variations on the northern continental slope are very different from the deep-sea region. We concluded that the high frequency BML variations on northern continental slope should be dominated by the tidal and near-inertial currents, and the low frequency BML variations in the deep-sea region may be controlled by dynamic processes, such as topographic Rossby waves or deep ocean eddies. To further confirm the results, more in situ observations, especially the direct observations of current velocities, are needed.

**Specific comments:**

1)  Section 1, Introduction: are there any previous papers discussing BML variations in the SCS or other regions?

Response: Thank you for your questions. In the revised manuscript, we added more relevant references and described the results from the references about the formation mechanisms of the BML and its variations, especially a recently published paper, entitled: "Spatial variation of bottom mixed layer in the South China Sea and a potential mechanism" by Li et al. (2022). In that paper, the authors investigated the thickness, stratification and spatial variation of the BML in the SCS,

and suggested that the mean value of thickness is about 154 m (based on 201 full depth profiles), which is much thicker than the mean values (73 m) as estimated in our analysis (based on 514 full depth profiles).

References:

Li, J., Yang, Q., Sun, H., Zhao, W., and Tian, J.: Spatial variation of bottom mixed layer in the South China Sea and a potential mechanism, Prog. Oceanogr., 206, 102856, https://doi.org/10.1016/j.pocean.2022.102856, 2022.

L85: "less than" should be replaced with "more than"?

Response: Thank you for pointing out this error in the manuscript, we have corrected.

2) L99: "where" should be replaced with "in the latter of which"?

Response: We have replaced "where" with "in the latter of which" in the revised manuscript.

3) L106-108: why high frequency signals are removed? I think BML is modulated with tidal periods.

Response: In order to explore the low frequency variations of bottom temperature, we use a 72-h low-pass filter to remove the inertial, tidal, and other high frequency signals from the raw data. But the power spectra are calculated from the raw data. To avoid possible misunderstanding, we have added a statement "the low-pass filter only used to explore the low frequency variations of temperature."

4) Section 2.3, The other dataset: the contents of the dissipation dataset should be described in more detail. What is the depth range of the two-dimensional dissipation map used in this study? What are "four dissipative processes" (L114-115)? What are "five process contributions" (L116)

Response: Thank you for your comment. We have revised the description of the dataset and added more detailed information. The modified sentences are as follows:

*The dataset consists of global column-integrated maps of internal tide energy sources and sinks with a horizontal resolution of 0.5° × 0.5°. In this dataset, energy sinks are provided for each of M2, S2 and K1 and for "All constituents" (eight most energetic tidal constituents). The energy sinks are decomposed into five process contributions: (i) dissipation of low modes via wave-wave interactions; (ii) dissipation of low modes scattering by abyssal hills; (iii) dissipation of low modes critical reflection; (iv) dissipation of low modes shoaling; (v) local dissipation of high modes. Units are Watts per square meter. Considering the dissipation of low modes via wave-wave interaction mainly occurs in the stratified water column "over" the BML, thus only the other four processes were used in this study.*

5) L124-125: I wonder if the relative variance method has systematic bias against the length of each CTD full-depth profile. Or bottommost partial profile data with some fixed length were used to avoid such bias?

Response: To avoid the systematic bias as mentioned by the reviewer, we adopt the bottommost partial profile with the specific length (usually twice the BML depth) to identify the BML. For some irregular profiles, we take a larger length to reduce the chance to mislocate the BML as suggested by Huang et al. (2018).

References:

Huang, P.-Q., Cen, X.-R., Lu, Y.-Z., Guo, S.-X., and Zhou, S.-Q.: An integrated method for determining the oceanic bottom mixed layer thickness based on WOCE potential temperature profiles, J. Atmos. Oceanic Technol., 35, 2289-2301, https://doi.org/10.1175/JTECH-D-18-0016.1, 2018.

6) L132: "higher" should be replaced with "highest" among the three H_BML estimates?

Response: Thank you for your suggestion. Although the quality index (QI) was used to evaluate the quality of the estimate of $H_{BML}$, we found some estimated $H_{BML}$ values with the highest QI may exist possible errors that may be due to contaminated samples or accidental spikes. To avoid the estimate of the $H_{BML}$ only by QI, all those profiles were visually inspected to detect these possible errors. In such a case, "higher" is more suitable than "highest".

7) L168-170: Why interesting? I wonder if the fitted curve equation (2) can be well approximated to R_H/D ~ 100/D, just indicating the BML thickness doesn't vary so much with the total depth.

Response: Based on the CTD observations, we found that the percentage of BML to the ocean depth ($R_{H/D}$) is roughly exponentially with a water depth that can be well fitted by equation (2). If we assume the $H_{BML}$ is constant (100 m), the fitted curve (red) by equation (2) is quite different from the relation curves (blue) calculated by constant $H_{BML}$ values (Figure R1). Considering the Reviewer's concern, we have replaced the word "It is interesting" with "In general" in the sentence.

[Figure]

Figure R1. The percentage of BML thickness to the water depth as a function of the water depth. Red line indicates the least-squares fit curve. Blue line is assumed that the constant BML thickness (100 m).

8) L187-L193: there are no descriptions of temporal variations of BML, especially unsteady features of thick MBL at M1, which should be pointed out here.

Response: Thank you for your suggestions. we have added the descriptions of temporal variations of BML in the revised manuscript.

9) L201-203: It is not clear to me how the maximum gradient of the first EOF mode is related to the quasi-homogeneous structure. I cannot see the clear merits of conducting the EOF analysis here.

Response: Considering the coarser resolution of mooring observations in the vertical direction and the linear interpolation method may induce some errors. We are not certain whether the vertical change of the first EOF modes is real exists at the bottom boundary or is induced by the linear interpolation method. The maximum gradient of the first EOF mode may not clear represent the quasi-homogeneous structure. So, we have deleted the EOF analysis in the revised manuscript.

10) Section 4, The potential formation mechanisms of the BML in the SCS: as described in my general comments, there are no discussions concerning the most relevant bottom Ekman layer dynamics.

Response: Thanks so much for your valuable suggestion. As answered in general comments, we have added some discussion before the mechanism analysis and narrowed our discussion focus on the BML differences between the northern continental slope and the deep-sea regions. The corresponding parts have been also revised in the revised manuscript.

11) L254: "Huang et al. (2019), which" should be replaced with "Huang et al. (2019) that"?

Response: We have modified the sentence as follows:

*These result shows no monotonic relationship between the $H_{BML}$ and the buoyancy frequency as suggested by Huang et al. (2019), in which they showed that the $H_{BML}$ tends to be thinner with stronger stratification.*

12) Figures 9c and 11: the best fit curve is empirical and doesn't necessarily have the wide applicability. Moreover, it should be noted that internal wave breaking occurs in the stratified part of the ocean above the BML, not in the BML. The obtained relationship between the BML thickness and internal tide dissipation should be regarded as secondary. Tidal flow itself and its frictional damping should play a primary role in the BML. Therefore, I think the authors' discussion is inappropriate.

Response: Thanks for your comment. We agree with the reviewer's comment that the best fit curve is empirical and its applicability needs further validation. Considering the Reviewer's concern, we

have added some statements that the interactions between currents and topography are very complex, thus more factors to affect the $H_{BML}$ should be considered in the future.

We totally agree with the reviewer's suggestion that the tidal flow itself and its frictional damping have an important role in the formation of BML. The nature of the BML is governed by the forcing of the boundary layer. We use the statistical relations ($H_{BML}$ and tidal dissipation) to predict the $H_{BML}$ in the SCS. It should be noted that the two-dimensional tidal dissipation dataset is depth-integrated and doesn't constant much information in the vertical direction. In this dataset, energy sinks are decomposed into five process contributions. Among these processes, the low-mode dissipation attributed to wave-wave interaction occurs in the stratified water column "over" the BML. Therefore, the dissipation of low modes via wave-wave interactions should be excluded from the total energy dissipation in this study. According to another reviewer's suggestions, we have modified the Figure 7 and corresponding parts in the revised manuscript.

---

## Author Comment (AC2)

Reply to reviews of "Spatial-temporal characteristics of the oceanic bottom mixed layer in the South China Sea"

**Reply to reviewer #2:**

We would like to thank you for your careful reading, helpful comments, and constructive suggestions, which have significantly improved the manuscript.

**Major comments**

1) Although the relative variance method is applied to potential temperature, salinity, and potential density measured with the CTD, the EOF analysis is applied only to potential temperature measured at the mooring sites. I understand that there are no salinity measurements at Station M1, but at least the salinity and potential density data at Station M2 should be analyzed to make sure that there exists no significant difference among the $H_{BML}$ estimated from potential temperature, salinity, and potential density.

Response: Thanks for your suggestion. Considering the EOF analysis maybe does not seem that significant to the main conclusions. Thus, we have deleted the EOF analysis contents in the revised manuscript. The explanations are also given in Comments 2 and 3.

2) The depths of sensors listed in Table 1 are much coarser than those of the circles in Figure 7. The authors should describe how to calculate the EOF modes more in detail.

Response: In the original manuscript, we use the linear interpolation method to interpolate the raw mooring data (seven sensors in each mooring) into 10 m intervals, and then conduct the Empirical Orthogonal Function (EOF) analysis. We consider the linear interpolation method may induce some spurious information, we have deleted the EOF analysis in the revised manuscript.

3) Figure 7a shows that stratification around 115 m is reduced (positive) when stratification in the BML (below 100 m) is enhanced (negative), and vice versa. By contrast, Figure 7b shows that stratification around 50 m is reduced (positive) when stratification in the BML is reduced (positive) as well. I feel the latter is physically reasonable, but the former is not. The authors

should explain that the former result is not inconsistent with the quasi-homogeneous structure shown in Figure 6.

Response: In the original manuscript, the inflection points around 115 m (Figure 7a) and 50 m (Figure 7b) can interpret as the maximum gradient. Considering the coarser resolution of mooring observations in vertical direction. We agree that the vertical structure of the first EOF modes may not be real and think that it could possibly result from the vertical interpolation. We have thus delected the the EOF analysis in the revised manuscript.

4) The total energy dissipation is not suitable for examining the influence of internal tide dissipation on $H_{BML}$ (Figures 9c and 10) because the low-mode dissipation attributed to wave-wave interaction occurs in the stratified water column "over" the BML. Instead, the sum of the "low_modes_shoaling", "low_modes_critical_slopes", "low_modes_scattering", and "high_modes_local" dissipation, all of which are included in the dataset provided by de Lavergne et al. (2019), should be used. The influence of external (barotropic) tide dissipation, which is not considered in the present study at all, should also be examined. In fact, Zu et al. (2008, Deep-Sea Res. Part I) demonstrated that the barotropic tidal energy is dissipated around the Luzon and Taiwan straits.

Response: According to the Reviewer' s suggestion, the wave-wave interaction dissipation has been excluded from the total energy dissipation, and we have recalculated the relations and modified the figures and corresponding parts in the revised manuscript.

It is a good suggestion to examine the influence of barotropic tidal energy on the distribution of the $H_{BML}$, which needs the energy dissipation near the near-bottom regions in the SCS can be well simulated. The internal tide energy sinks datasets used in this study include the barotropic and baroclinic tidal energy, but the depth-integrated internal wave energy dissipation doesn't contain much information in the vertical direction. Further studies are needed to address this issue using a more complete and high-resolution ocean model. We will leave this for our further study.

5) Eq. (3) is valid only when the vertical eddy diffusivity, Az, is constant. In the present case that Az varies in the vertical direction, the right-hand side of Eq. (3) must be d(Av(dT/dz))/dz. The relevant parts of the manuscript should be revised as well.

Response: We are very grateful for your comments. Based on the mooring observations, we estimated three terms of $\partial T/\partial t$, $\partial T/\partial z$, and $\partial^2 T/\partial z^2$ in Eq. (3), then the unknown values of $w$ and $A_z$ can be estimated from a set of linear equations using the least-square fitting method. So the calcucated $w$ and $A_z$ are depth-averaged values. Considering the Reviewer' s concern, we have added the explations that the vertical eddy diffusivity, $A_z$, is constant at the vertical observation range of two moorings.

**Minor comments**

6) Line 85: "less than 50 m" should read "more than 50 m".

Response: Thank you for pointing out this error in the manuscript, we have corrected.

7) Lines 191-192, Figure 6: This is not a fair comparison because the contour interval of Figure 6a(6b) is ten times larger than that of Figure 6c(6d).

Response: The ranges of potential temperature at M1 and M2 were about 2.2~2.4°C and 2.06~2.08°C, respectively. The variations of the potential temperature difference at M1 was 10 times greater than M2. For a fair comparison, we choose a normalized value, 25% variations of the potential temperature gradients, as the index for comparison. This value is $-4\times10^{-4}$ °C m$^{-1}$ at M1 and $-0.25\times10^{-4}$ °C m$^{-1}$ at M2.

8) Lines 106-108, Figure 8: The power spectra shown in Figure 8 cannot be calculated from the filtered data where the inertial, diurnal, and semidiurnal signals are removed. To avoid misunderstanding, the authors should mention that the power spectra are calculated from the raw data.

Response: According to the Reviewer' s suggestion. We have added the contents to mention that the power spectra are calculated from calculated from the hourly mean data.

9) Lines 212-214, Figure 8: For the benefit of a reader, the heights of the data used for spectral analysis should be denoted.

Response: We have added the depth information in the figure.

10) Figure 2: The title of the y-axis is missing.

Response: Thank you so much for your careful check. We have added the title of the y-axis in Figure 2 in the revised manuscript.

11) Figure 6: For the benefit of a reader, the depths of sensors should be superposed.

Response: We have marked the depths of sensors with "☆" in Figure 6.